# Microstructure and Corrosive Wear Properties of CoCrFeNiMn High-Entropy Alloy Coatings

**DOI:** 10.3390/ma16010055

**Published:** 2022-12-21

**Authors:** Haodong Wang, Jiajie Kang, Wen Yue, Guo Jin, Runjie Li, Yongkuan Zhou, Jian Liang, Yuyun Yang

**Affiliations:** 1School of Engineering and Technology, China University of Geosciences (Beijing), Beijing 100083, China; 2Zhengzhou Institute, China University of Geosciences (Beijing), Zhengzhou 450001, China; 3College of Material Science and Chemical Engineering, Harbin Engineering University, Harbin 150001, China; 4Institute of Exploration Techniques, Chinese Academy of Geological Sciences, Langfang 065000, China

**Keywords:** cold spraying, high-velocity oxygen fuel spraying, high-entropy alloy, wear, corrosion

## Abstract

In order to improve the wear resistance of offshore drilling equipment, CoCrFeNiMn high-entropy alloy coatings were prepared by cold spraying (CS) and high-speed oxygen fuel spraying (HVOF), and the coatings were subjected to vacuum heat treatment at different temperatures (500 °C, 700 °C and 900 °C). The friction and wear experiments of the coatings before and after vacuum heat treatment were carried out in simulated seawater drilling fluid. The results show that CoCrFeNiMn high-entropy alloy coatings prepared by CS and HVOF have dense structure and bond well with the substrate. After vacuum heat treatment, the main peaks of all oriented FCC phases are broadened and the peak strength is obviously enhanced. The two types of coatings achieve maximum hardness after vacuum heat treatment at 500 °C; the Vickers microhardness of CS-500 °C and HVOF-500 °C are 487.6 and 352.4 HV_0.1_, respectively. The wear rates of the two coatings at room temperature are very close. CS and HVOF coatings both have the lowest wear rate after vacuum heat treatment at 500 °C. The CS-500 °C coating has the lowest wear rate of 0.2152 mm^3^ m^−1^ N^−1^, about 4/5 (0.2651 mm^3^ m^−1^ N^−1^) of the HVOF-500 °C coating. The wear rates and wear amounts of the two coatings heat-treated at 700 °C and 900 °C decrease due to the decrease in microhardness. The wear mechanisms of the coatings before and after vacuum heat treatment are adhesive wear, abrasive wear, fatigue wear and oxidation wear.

## 1. Introduction

Compared with land drilling, offshore drilling is more difficult and riskier. In the process of deep-sea drilling, the drill bit not only needs to withstand the corrosive media such as chloride ions in seawater and drilling fluid, but also needs to improve its wear resistance to prevent rock cuttings impact during drilling. Therefore, the preparation of high-wear-resistance coatings on drill pipes has important economic significance and engineering application value to ensure the safety of drilling equipment, improve drilling efficiency and control costs.

In 2004, Yeh first proposed a new multialloy design concept: high-entropy alloys (HEAs), breaking the traditional design strategy of one or two elements alloying. Compared with traditional alloys, high entropy effect, lattice distortion, diffusion retardation and cocktail effect are the most significant effects in high-entropy alloys [1,2]. These effects can provide several important advantages, including easy access to supersaturated state and fine precipitates; increasing recrystallization temperature; slowing down grain growth and particle coarsening rate, which may be beneficial to its mechanical properties; excellent thermal stability; and good wear resistance and corrosion resistance [3,4,5].

The atomic radii of Co, Cr, Fe and Ni are 0.126, 0.127, 0.127, 0.124 nm, respectively. They easily form FCC structures, and CoCrFeNi-based alloys are common combinations in high-entropy alloys. As one of the simplest structures, CoCrFeNiMn high-entropy alloys with FCC structures have been widely studied [6,7]. Heat treatment promotes the uniformity of phase and grain size in the alloy without changing the proportion of the original elements, thereby improving mechanical properties [8,9,10]. Stepanov et al. [11] annealed the CoCrFeNiMn high-entropy alloy at 600–800 °C for 1 h, and Cr-rich second phases with tetragonal (sigma phase) and bcc structures were formed in the alloy. Liu et al. conducted heat treatment on CoCrFeMnNi high-entropy alloy at 700 °C, and the alloy produced recrystallization and precipitation of fine σ particles, which improved the yield strength and tensile property of the alloy [12].

Heat treatment of high-entropy alloy can not only improve its mechanical properties, but also improve its wear resistance [13,14,15]. The FeCoCrNiMnAl high-entropy alloy cladding layer was prepared by laser cladding technology. After heat treatment at 600 °C, the wear resistance of FeCoCrNiMnAl coating decreased by 11.7% [16]. Guo et al. studied the effect of heat treatment on wear behavior of AlCoCrFeNiSi high-entropy alloy, and the results showed that after heat treatment at 1150 °C, the friction coefficient of high-entropy alloy decreases significantly (0.115) [17]. There are also some studies on the effect of heat treatment on the wear behavior of high-entropy alloy in corrosive solution. The wear behavior of AlCoCrFeNiCu high-entropy alloy in hydrogen peroxide solution was influenced by the heat treatment temperature, and with the increase in heat treatment temperature (RT, 600, 700, 1000 °C), the average friction coefficient of the alloy increased from 0.037 to 0.115, and the wear amount increased from 3µm to 11µm [18]. Yu et al. studied the effect of heat treatment on the wear behavior of AlCoCrFeNi (Ti_0.5_) high-entropy alloy under oil and MACs lubrication. After heat treatment, AlCoCrFeNi alloy preserved an equiaxed crystal structure, AlCoCrFeNiTi_0.5_ alloy kept a dendrite structure, and both alloys had excellent wear resistance [19].

High-velocity oxygen fuel spraying technology has higher particle velocity, lower flame temperature and fewer unmelted particles than atmospheric plasma spraying, which can prepare anticorrosive and wear-resistant coatings with low oxidation degree, good bonding rate and low porosity [20,21]. Cold spraying technology relative to the traditional thermal spraying technology, as well as the use of less thermal energy and more kinetic energy, greatly improves the impact speed of spraying particles on the substrate surface, and still keeps them in a solid state [22,23]. Nikbakht et al. selected equiatomic CoCrFeNiMn high-entropy alloy powders to study the particle-bonding mechanism during cold spray deposition [24]. The results show that the impact morphology of particles is related to strain rate, in addition to material and microstructure. Due to the dynamic inertial effect, severe plastic deformation occurred in the lower half of the particles, and deformed nanotwins with a thickness of about 200 nm were observed. Ahn et al. also prepared equiatomic CoCrFeNiMn high-entropy alloy coatings by a cold spraying process, and studied the effect of the coating microstructure on the nanoindentation performance [25]. It was found that the grain size of the sprayed coating is fine, and there are a large number of twins in the powder particles, which is also the reason for the excellent performance of the CoCrFeNiMn high-entropy alloy coating. Silvello et al. systematically evaluated the effects of CS and HVOF on the corrosion resistance and wear resistance of CoCrFeNiMn high-entropy alloys [26]. Both the two spraying techniques could prepare dense and thick alloy coatings with relatively low porosity. The results showed that no phase transformation was detected in the CS coating, while the oxides of Fe, Mn and Cr could be observed in the HVOF coating. The corrosion current and wear rate of the CS coating were 1.2 μA and 2.80 × 10^−4^ mm^3^ N^−1^ m^−1^. Moreover, the corrosion current and wear rate of the HVOF coating were 0.29 μA and 6.70 × 10^−5^ mm^3^ N^−1^ m^−1^. Both of them showed good wear resistance and corrosion resistance.

Although there have been many reports on the wear and mechanical properties of high-entropy alloys after heat treatment, there are few studies on the wear behavior of CoCrFeNiMn high-entropy alloy coating after heat treatment, especially under simulated seawater drilling fluid. In this study, CoCrFeNiMn high-entropy alloy coatings were prepared by cold spraying and high-velocity oxy-fuel spraying. The effects of heat treatment temperature (500 °C, 700 °C, 900 °C) on the phase composition of high-entropy alloy coatings were studied. The friction and wear experiments under simulated seawater drilling fluid immersion were carried out to compare the differences in wear mechanism and corrosion wear properties.

## 2. Experimental Section

In this study, high-purity bulk iron, cobalt, chromium, nickel and manganese (purity > 99.99%, produced by BGRIMM Technology group, Beijing, China) were selected as raw material. The bulk metal was heated in a crucible according to a certain atomic ratio, and N_2_ was used to protect the melting process. The molten metal liquid was formed into fine liquid particles by atomizing nozzle under vacuum environment and cooled to obtain high-entropy alloy powders. The specific nominal composition is shown in Table 1.

The high-entropy alloy powders with the particle size distribution of 30~60 μm and 10~30 μm were selected for the preparation of coatings by supersonic flame spraying and cold spraying technology, respectively. The substate material was 35CrMo alloy structural steel, which is widely used in the drilling field because of its good static strength, impact toughness and high fatigue limit. The size of the block sample was processed into 50 mm × 15 mm × 5 mm by wire electrical discharge machining. The 35CrMo substrate was degreased and sandblasted in order to obtain better mechanical bonding strength and appropriate surface roughness. Sandblasting material was 200-mesh-sieved emery sand, sandblasting pressure was 0.6 bar, angle was 90° and sandblasting distance was 30 mm. In addition, the surface of the sandblasted sample was further cleaned with an air ejector to remove debris from the substrate surface. The cold spraying equipment (Impact company, Bavaria, Germany, Impact 5/11) and high-velocity oxygen fuel spraying equipment (Kermetico company, Benicia, CA, USA, gun model AK-06) were used to prepare coatings. HVOF high-entropy alloy coatings with the thickness of about 250 μm and CS high-entropy alloy coatings with the thickness of about 300 μm were prepared on the substrate surface using hydrogen as fuel. The detailed process parameters of CS and HVOF are shown in Table 2.

The heat-treatment environment of CoCrFeNiMn high-entropy alloy coatings prepared by two kinds of spraying technology is vacuum, and the heat treatment is carried out in HTVO-1200 vacuum annealing furnace at a pressure of less than 10^−4^ Pa. The heat treatment temperature is set to 500 °C, 700 °C and 900 °C. The heating rate is 10 °C/min and heat preservation is 2 h. Finally, the samples were cooled to room temperature in furnace.

Two kinds of CoCrFeNiMn high-entropy alloy coatings and the coatings after heat treatment at different temperatures were tested by CETR-UMT-3 multifunctional wear tester. The friction and wear experiments were carried out by soaking the samples in simulated seawater drilling fluid as corrosive medium. Before the experiment, the coating surfaces were polished. The coupled ball is made of Φ6 silicon nitride, setting the normal load of 10 N, frequency of 5 Hz, test time of 1800 s, and stroke of 6 mm. Each coating sample was repeated for 3 times and averaged to increase the reliability of the experiment. The effects of heat treatment at different temperatures on the corrosive wear properties of CS and HVOF high-entropy alloy coatings were investigated, and the coupling wear mechanism between corrosion and friction was studied. The wear rate is characterized by the volume loss before and after the wear experiment. The wear rate calculation formula is as follows:(1)$$Q=\frac{V_{w}}{NS}$$
where *Q* is the wear rate (mm^3^·N^−1^·m^−1^); *V_w_* is the wear volume(mm^3^) obtained from the contour integral of the wear scar cross sections; *N* and *S* represent the applied load (*N*) and the total travel distance (mm), respectively.

The surface and cross-section morphologies of CoCrFeNiMn high-entropy alloy powders and coatings before and after heat treatment were observed by MERLIN Compact scanning electron microscope (SEM). The porosity of the coating was calculated by using the software ImageJ2X to perform gray processing on the SEM image of the coating cross section and averaging the multiple (≤10 times) selection measurements. The three-dimensional morphologies of the coatings were measured by a three-dimensional white light interference surface profiler (NexView, ZYGO, Middlefield, CT, USA). The grain size, shape and orientation of the two kinds of CoCrFeNiMn high-entropy alloy coatings were characterized by OXFORD SYMMETRY EBSD electron backscatter instrument from Oxford Instruments. The phase composition of the two kinds of high-entropy alloy powders, original coatings and coatings after heat treatment at different temperatures were analyzed by X-ray diffractometer (GE, Germany) with Cu Kα target. The diffraction angle was set in the range of 2*θ* = 20~100°, and the scanning speed was 4°/min. The microstructure of the coating was observed by Talos F200X G2 transmission electron microscope with the accelerating voltage of 300 kV. Double spray thinning was performed in a mixed solution of HClO_4_:C_2_H_6_O = 1:9 at about −30 °C. The sample was further thinned by a Gatan Duo-Mill argon ion mill with 5 kV Ar ions at an angle of 5°. The microhardness of two kinds of CoCrFeNiMn high-entropy alloy coatings and the coatings after heat treatment at different temperatures were measured by microhardness tester (MICROMET-6030, Buehler, Ratingen, Germany). The load was 1 N and the duration was 10 s. Five points were measured and the average hardness was calculated. In addition, in order to make the measured value accurate and effective, the coating defect positions were avoided when taking points.

## 3. Results and Discussion

### 3.1. Phase Structure and Morphology of CoCrFeNiMn High-Entropy Alloy Coatings

Figure 1 shows the microstructure of CoCrFeNiMn high-entropy alloy powders prepared by vacuum atomization (CS and HVOF raw material powders). The powders prepared by gas atomization have a good spherical and smooth surface, as shown in Figure 1a,b. The diameter of CS powders is about 10~30μm. HVOF raw material powders have a diameter of about 30–60 μm, and the powder consists of a small portion of ultrafine particles with a particle size of less than 30 μm, proving it can not remove all extremely fine particles by sieving [27]. The enlarged SEM secondary electron images show that there are dendrites on the surface of the high-entropy alloy powders, and several small satellite balls are attached to the surface of the coarse particles, which is caused by the collision of turbulence and condensed droplets in the gas atomization chamber (Figure 1c,d) [28]. This spherical morphology promotes the fluidity of the powders, improves the delivery efficiency of the spray gun, and facilitates powder feeding during CS and HVOF spraying. In addition, it also includes nonspherical shapes, which may be formed by the coverage of large particles and small particles.

Figure 2 shows the XRD patterns of CoCrFeNiMn high-entropy alloy powders. The powder has three main peaks at (111), (200) and (220), which appear near 2θ = 43.7°, 51.6° and 74.5°, respectively. This is the characteristic of single-phase FCC. In addition, the non-equilibrium state during vacuum atomization is another factor in the formation of a simple solid solution phase [29]. It shows that the element distribution of CoCrFeNiMn high-entropy alloy powders prepared by vacuum atomization is relatively uniform, which confirms that the powder composition is very close to the atomic composition.

Figure 3 shows the XRD patterns of CoCrFeNiMn high-entropy alloy coatings prepared by CS and HVOF without heat treatment. The results show that the crystal structure of CoCrFeNiMn high-entropy alloy powders is still FCC phase after coating, which is consistent with that of the powders. The peak value of FCC phase in the HVOF coating is higher than that of the CS coating. It is worth noting that the peak intensity of the CS coating changes due to the lattice distortion caused by different atomic sizes, but still keeping the FCC structure without phase transition. However, some new peaks of Fe-rich and Cr-rich oxides appeared in HVOF coatings at 2θ = 25.6° and 30.2°, indicating that partial oxidation occurred during thermal spraying.

Figure 4 shows the XRD patterns of CoCrFeNiMn high-entropy alloy coatings prepared by CS and HVOF after heat treatment at different temperatures. The results show that the main peaks of all oriented FCC phases are broadened and the strength is significantly enhanced after heat treatment for both CS and HVOF alloy coatings. This is mainly due to the grain refinement, unfavorable crystallinity and lattice distortion caused by heat treatment. In addition, as shown in Figure 4a, the collision between the high-entropy alloy raw material powders and the matrix is unavoidable during the CS process, resulting in severe plastic deformation and high strain rate of the alloy powders, and the transformation from FCC phase to FCC phase and B_2_ phase occurs. As shown in Figure 4b, the HVOF coating produces spinel and Cr_2_O_3_ phases after heat treatment. In addition, the peaks of Fe and Mn oxides continue to increase as the heat-treatment temperature increases. Among them, 500 °C is the temperature point at which recrystallization nucleation begins, so some nanocrystals are produced, but the grains do not grow at this temperature, resulting in FCC peak broadening [30].

Figure 5a,b are the cross-sectional morphologies of the CS coatings. The middle of two red dotted lines is the coating. The coating is well-bonded to the substrate, and the FCC phase of the CoCrFeNiMn high-entropy alloy is easily deformed [31,32]. Due to severe plastic deformation of powders, high hardness and high strength can be obtained [33,34]. In addition, dense coatings can be obtained. The coating cross section was selected multiple times by ImageJ2x software. The coating porosity was about 1.1% after gray processing. There are a small number of cracks and pits in the coating, which may be due to the decrease in velocity and strain rate when the alloy powders collided with the substrate. Figure 5c,d is the cross-sectional morphology of the HVOF coating. The powder has a good degree of melting, and the porosity is about 1.4%. Compared with that of the CS coating, the porosity is higher, and some pores or microcracks formed in the area where the metal connection is not completed. These areas may be the separation position of the melting zone and the unmelted zone.

Figure 6 shows the microhardness distribution curves of the CS coating and HVOF coating along the cross section. The average hardness of the CoCrFeNiMn high-entropy alloy coating prepared by CS is 317.1 HV_0.1_, and the maximum hardness is 324.7 HV_0.1_, which is higher than that of the substrate and HVOF coating. In addition, the CS coating has better bonding strength with the substrate. The increase in the hardness of the matrix is due to the increased bonding strength during blasting.

Figure 7 shows the microhardness distribution curves of CoCrFeNiMn high-entropy alloy coatings prepared by CS and HVOF, and the coatings heat-treated at different temperatures. Both coatings reach the highest hardness at 500 °C, 487.6 HV_0.1_ for CS coating and 352.4 HV_0.1_ for HVOF coating. At 700 °C and 900 °C, the hardness of both coatings is lower than that at room temperature. This is because the heat treatment at 500 °C makes the coating form a heterogeneous structure composed of nanotwins in the recrystallized grains, and high-density dislocations appear in the deformation region. When the annealing temperature exceeds 500 °C, the coating exhibits a completely recrystallized structure, the yield stress decreases and the hardness decreases.

As shown in Figure 8, the EDS mapping of the coatings (colored right images) is obtained based on the middle images. The surface of CS and HVOF coatings after vacuum heat treatment at 500 °C is relatively flat, which is consistent with that at room temperature. After vacuum heat treatment at 700 °C, the microstructure of the coating recovers and recrystallizes, resulting in a dendritic macrostructure of the coating [35]. At 900 °C, the enrichment of the coating surface is more obvious. Due to the different components, the dendritic core region is darker than the dendritic region, and the difference between the two regions comes from the coring effect. The microstructure consists of a dendritic core zone rich in Ni and Fe and a dendritic zone rich in Cr and Mn. The microstructures of both regions are composed of FCC and B_2_ precipitates. The microstructure of the dendritic core region is composed of B_2_ precipitates and BCC precipitates with eutectic structure, while the microstructure of the dendritic region is composed of FCC matrix and B_2_ precipitates [36].

Figure 9 shows the results of phase analysis of CS and HVOF coatings by electron backscatter diffraction (EBSD) orientation microscope. Figure 9a,b are the corresponding SEM images of the cross section of the CS coatings, and Figure 9c,d are the corresponding SEM images of the cross section of the HVOF coatings. The BCC phase of the CS and HVOF coatings is Ni-rich. The dislocation density and stress of high-entropy alloys correspond to a lower proportion of indexless black spots. Compared with the HVOF coating, the high degree of deformation in the CS coating significantly reduces the Index quality of the IPF map,”from’76% of the points to be indexed, which is lower than the 92% index quality of HVOF. The relatively small grain size of the CS coating may be due to the excessive movement and interaction of dislocations during dynamic loading. CS treatment has an obvious refining effect on the grains of CoCrFeNiMn high-entropy alloy, and the particles are seriously deformed due to the dynamic inertia effect and adiabatic deformation during impact. At the same time as the high-strain plastic deformation, the dislocations at the crystal interface increase. The dislocation density also increases to form dislocation cells, and the coarse grains are refined into subgrains. In the local interface region where the maximum plastic deformation occurs in the granular material, dynamic recrystallization occurs under the combined action of adiabatic heating and plastic deformation, which further refines the subgrains into ultrafine grains [37,38]. The grain of HVOF coating is relatively large, and annealing twins will appear after heat treatment. However, even coarse grains have irregular shapes and many curved boundaries. With the increase in heat treatment temperature, both the size of recrystallized grains and the speed of microstructure coarsening increase, resulting in a more uniform microstructure and an increase in the proportion of twin boundaries [39]. Due to the good thermal stability of CoCrFeNiMn high-entropy alloy, the HVOF coating retains most of the complete grains, and the grains are relatively close to each other.

Figure 10 are TEM-selected area diffraction patterns of CS and HVOF high-entropy alloy coatings. No phase transition occurs for the CS coating, and it is a typical FCC phase. Selection of crystalline phase and twins along the band axis [011] and [0-1-1] electron diffraction patterns proved the existence of a twin structure (Figure 10c,d).

Figure 11 shows the TEM micromorphology of CS and HVOF high-entropy alloy coatings. FCC and B_2_ phases exist in the CS coating, while the second phase, except FCC, also exists in the HVOF coating (Figure 11a). The HVOF coating has an FCC twin structure and a large number of shear bands caused by stress at the twins (Figure 11b). Part of the twin structure existing in FCC interacts to form a cross structure, and the cross effect may affect the mechanical properties of high-entropy alloys.

### 3.2. Friction and Wear Properties of CoCrFeNiMn High-Entropy Alloy Coating at Different Temperatures

Figure 12 shows the friction coefficient curves of CoCrFeNiMn high-entropy alloy coatings prepared by CS and HVOF and those of coatings heat-treated at different temperatures. The friction coefficients of the CS coating and HVOF coating are relatively stable. The friction coefficients of the RT and 500 °C heat-treated CS coatings are relatively low in the first 200 s, which is due to the fact that the oxide film (Such as Cr_2_O_3_ and CoO) on the CS coating is relatively dense below 500 °C [16,26], and was not destroyed by the precipitates of high-temperature phase, and the passivation film inhibits the corrosion behavior of Cl^-^. The friction coefficients of the RT, 500 °C and 700 °C heat-treated HVOF coatings are very similar. In the 500 s friction and wear experiment, the friction coefficient of the 900 °C heat-treated HVOF coating has a significant upward trend, which may be due to the relatively low hardness of the coating. It can be seen from Figure 12 that the curve can be divided into two stages: the running-in stage and stable stage. Among them, the COF value of the CS-500 coating changes most obviously. It increases sharply in the running-in stage, and then decreases rapidly to the steady-state value. The reason is that the surface roughness is different, and the friction process is constantly adjusted with the change in contact surface [40]. During the running-in period, the asperities on the contact surface were crushed and the contact area increased. During the stable period, a dynamic balance was formed between the formation and removal of the oxide layer, and the friction coefficient fluctuated within a certain range. The rapid decrease after the initial peak came from the formation of low shear transfer film during friction. The surface hardness of the two coatings after heat treatment at 700 °C and 900 °C decreased due to the generation of soft FCC matrix and B2 precipitates on the surface and the rupture of the oxide layer. This made the coating unable to resist the normal load and shear force of the coupled part at the beginning, resulting in a relatively stable COF value. In 900 s, due to the generation of a large number of holes on the surface of HVOF-900 °C, the COF value continued to increase, which was the main reason for the largest cross-sectional volume of the wear scar. The holes made the tiny particles in the drilling fluid cause more serious three-body wear on the coating, which was also consistent with the characterization of the microscopic morphology of the wear scar in the next part. In addition, the friction coefficient of the substrate was about 0.35, which is higher than that of the coatings.

Figure 13 shows the wear cross-sectional volumes of CoCrFeNiMn high-entropy alloy coatings prepared by CS and HVOF and heat-treated coatings at different temperatures. The wear scar volume of the two coatings basically changes with the change in hardness. After heat treatment at 500 °C, the hardness of the two coatings reached the maximum, and the wear scar volume was relatively small. The wear depth of the CS coating was about -6.8μm, while that of the HVOF coating was about -8μm. In the comparison of the wear depths of the 900 °C heat-treated coatings, the wear degree of the CS coating is also lower than that of the HVOF coating.

Figure 14 shows the wear amount and wear rate of CS and HVOF coatings heat treated at different temperatures. In general, the variations of the two coatings are relatively consistent. The wear rates of the two coatings without heat treatment are very close, which are 0.2952 mm^3^ m^−1^ N^−1^ and 0.3016 mm^3^ m^−1^ N^−1^, respectively. The wear rate of the CS-500 °C coating is 0.2152 mm^3^ m^−1^ N^−1^, which is about 4/5 (0.2651 mm^3^ m^−1^ N^−1^) of that of HVOF-500 °C coating. The sliding wear resistance of the CS coating is better compared to the HVOF coating. After heat treatment at 500 °C, the hardness of the HVOF coating and CS coating increases and the internal stress decreases, so the wear resistance is better. High-temperature heat treatment (700, 900 °C) leads to grain coarsening, thus reducing hardness and wear resistance. The wear rate of the HVOF-900 °C coating is the highest, about 1.1 times that of the CS-900 °C coating. The wear volume and wear rate of the substrate are 0.0275 mm^3^ and 0.4534 mm^3^ m^−1^ N^−1^, respectively. This also shows that the coatings can protect the substrate after reasonable heat treatment.

In order to analyze the wear mechanism of CoCrFeNiMn high-entropy alloy coatings prepared by CS and HVOF and heat-treated coatings at different temperatures, the wear morphology was characterized by SEM and EDS. Figure 15 shows the SEM wear morphology and EDS spectrum analysis of CS coatings under different heat-treatment conditions. The EDS mapping of the coatings (colored right images) is obtained based on the third image per row. The middle of two yellow dotted lines is the wear region. There are a large number of furrows on the wear surface of CS coating at room temperature, and the depth of furrows is uneven (Figure 15b,c). A small amount of enrichment of Si element can also be seen in EDS, and the enrichment area is consistent with the position of deep furrows in SEM. This indicates that the furrows on the wear surface of the coating were caused by the abrasive wear between the coating and the counter-wear ball and the three-body wear between the counter-wear ball and the quartz sand in the drilling fluid. In addition, there were also peeling pits on the surface of the coating, indicating the occurrence of fatigue wear. This is due to the fact that the counter-wear ball was pressed into the coating during the friction and wear experiment to make the coating subject to complex alternating stress loads and the brittleness of the coating material itself [39]. After vacuum heat treatment at 500 °C, the hardness of the CS coating increased due to a small amount of FCC phase, grain refinement, lattice constant increase and lattice distortion (Figure 15f,g). The area of deep furrow decreased significantly, but the three-body wear area still existed. The partial oxidation area on the wear surface of the coating is highly coincident with the Si element enrichment area, which may be due to the cracks and spalling of the coating caused by the quartz sand in the drilling fluid during the friction and wear experiment. For the room-temperature coating, the high hardness of the coating can resist some abrasive wear and fatigue wear. However, after vacuum heat treatment at 700 °C and 900 °C, due to the decreased hardness of the CS coating, the abrasive wear of the counter-wear ball and quartz sand on the coating surface was intensified, and the crack expansion led to the peeling of the coating.

Figure 16 shows the SEM wear morphology and EDS spectrum analysis of HVOF coating under different heat-treatment conditions. The middle of two red dotted lines is the wear region. The three-body wear between the coating, counter-wear ball and quartz sand in the drilling fluid is still the main wear mechanism. Since the HVOF coating is a traditional layered structure of thermal spraying, the peeling phenomenon is more obvious than that of the CS coating. The EDS results in Figure 16d show that there are also enrichments of O and Si elements in addition to Fe and Cr elements, indicating that the metal elements in the coating, such as Fe and Cr, formed an oxide layer after vacuum heat treatment. With the change in sliding conditions, the contact temperature of the friction surface, i.e., the flash point temperature, increased, which led to oxidative wear and adhesive wear [39]. With the increase in vacuum heat-treatment temperature, the coating produced a low-temperature hard solid solution, the crack width and spalling volume were reduced and the fatigue wear was alleviated, but the abrasive wear still appeared. With the increase in vacuum heat-treatment temperature, the oxidation wear of the HVOF coating after vacuum heat treatment at 700 °C and 900 °C was more obvious. The crack width and spalling volume increased, and the oxide layer grew, broke and peeled periodically, leading to higher wear rate.

In summary, the wear mechanisms of CS and HVOF coatings before and after vacuum heat treatment are adhesive wear, abrasive wear, fatigue wear and oxidation wear. The CS coating is mainly abrasive wear, and the HVOF coating is mainly oxidative wear, adhesive wear and fatigue wear due to the formation of oxides and layered structure during vacuum heat treatment.

## 4. Conclusions

The CoCrFeNiMn high-entropy alloy coatings were prepared by cold spraying and high-velocity oxy-fuel spraying. The effects of heat treatment on the phase composition, microhardness and wear behavior of high-entropy alloy coatings were studied. The main results are as follows:(1)CoCrFeNiMn high-entropy alloy powders prepared by gas atomization for CS and HVOF have good spherical shape, smooth surface and dendritic structure. The CoCrFeNiMn high-entropy alloy coatings prepared by CS and HVOF have dense structures and bond well with the substrate. They are all single-FCC solid solution structures, and the porosity is less than 1.5%. Compared with the CS coating, the HVOF coating has higher porosity.(2)After heat treatment, the main peaks of all oriented FCC phases are broadened and the strength is obviously enhanced. Both coatings reach the maximum hardness after vacuum heat treatment at 500 °C, and the Vickers microhardness of CS-500 °C and HVOF-500 °C are 487.6 and 352.4 HV_0.1_, respectively.(3)The wear rates of the two coatings at room temperature are very close. The wear rate for the CS and HVOF coatings reaches its the lowest point after vacuum heat treatment at 500 °C. The wear rate of the CS-500 °C coating is 0.2152 mm^3^ m^−1^ N^−1^, which is about 4/5 (0.2651 mm^3^ m^−1^ N^−1^) of that of the HVOF-500 °C coating. The CS coating with 500 °C vacuum heat treatment has the best wear resistance due to having the highest microhardness. The wear rates and wear amounts of the two coatings heat-treated at 700 °C and 900 °C decreases due to the decrease in microhardness. The wear mechanisms of the CS and HVOF coatings before and after vacuum heat treatment are adhesive wear, abrasive wear, fatigue wear and oxidation wear.

## Figures and Tables

**Figure 1 materials-16-00055-f001:**
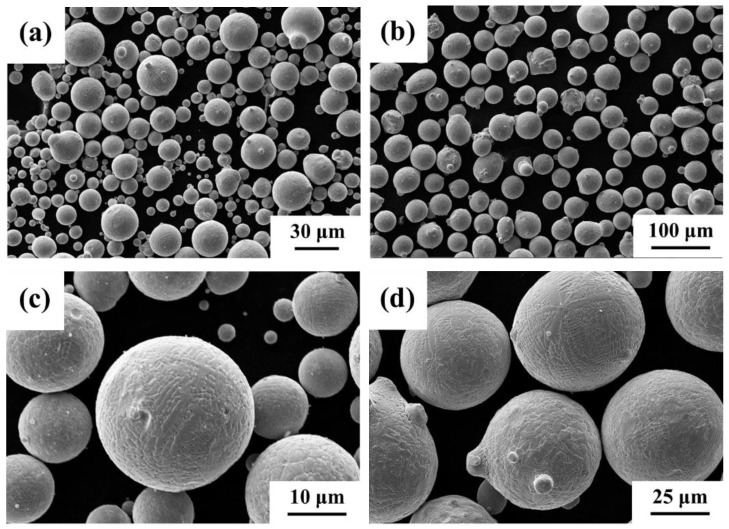
SEM secondary electron images of CoCrFeNiMn high-entropy alloy powders. (**a**,**c**) CS powders; (**b**,**d**) HVOF powders.

**Figure 2 materials-16-00055-f002:**
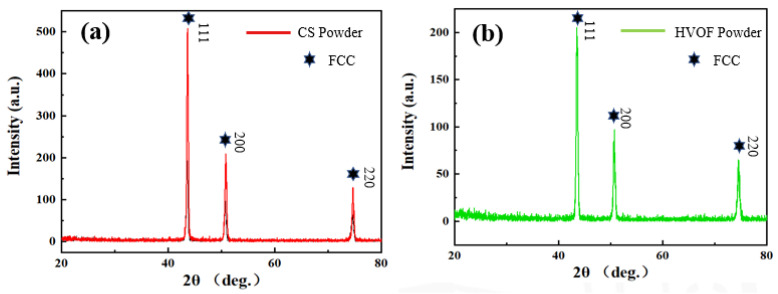
XRD patterns of CoCrFeNiMn high-entropy alloy powders. (**a**) CS powders; (**b**) HVOF powders.

**Figure 3 materials-16-00055-f003:**
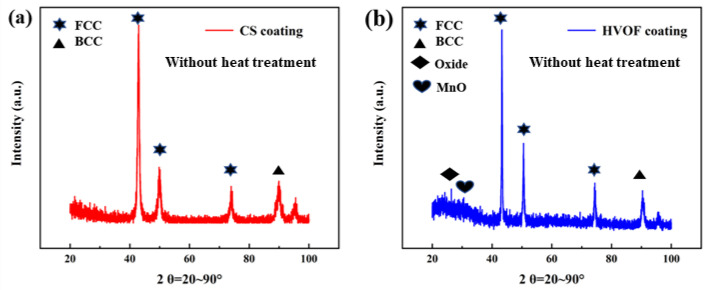
XRD patterns of CoCrFeNiMn high-entropy alloy coatings prepared by CS and HVOF without heat treatment. (**a**) CS coating; (**b**) HOVF coating.

**Figure 4 materials-16-00055-f004:**
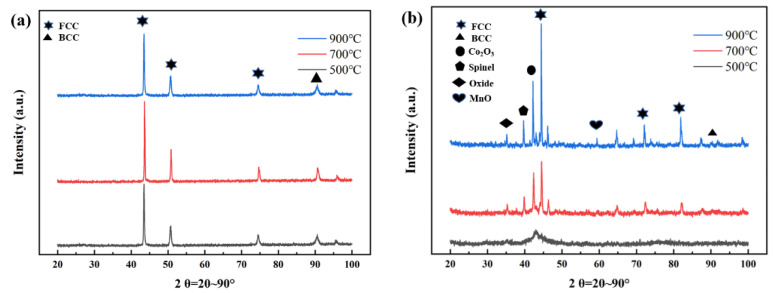
XRD patterns of CoCrFeNiMn high-entropy alloy coatings prepared by CS and HVOF after heat treatment at different temperatures. (**a**) CS coating; (**b**) HOVF coating.

**Figure 5 materials-16-00055-f005:**
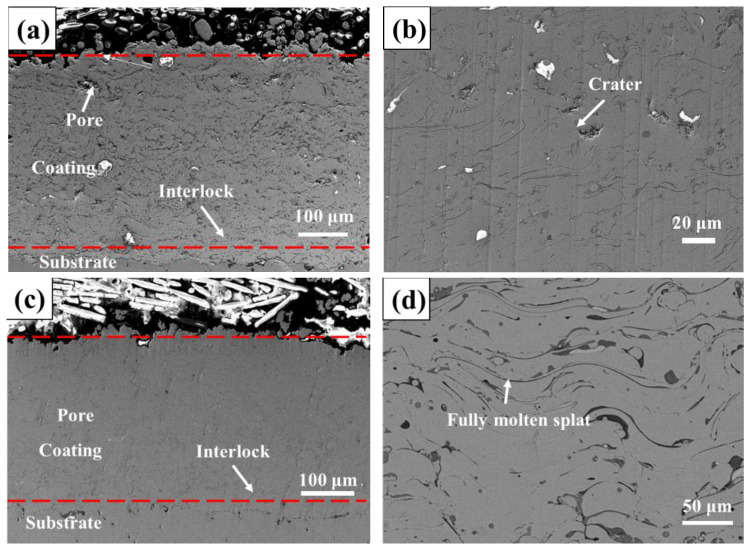
SEM images of high-entropy alloy coatings. (**a**) Low-magnification cross-section image of CS coating; (**b**) high-magnification cross-section image of CS coating; (**c**) low-magnification cross-section image of HVOF coating; (**d**) high-magnification cross-section image of HVOF coating.

**Figure 6 materials-16-00055-f006:**
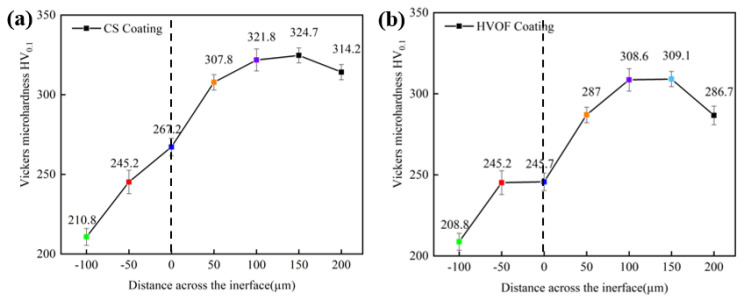
Microhardness–depth curves of high-entropy alloy coatings. (**a**) CS coating; (**b**) HOVF coating.

**Figure 7 materials-16-00055-f007:**
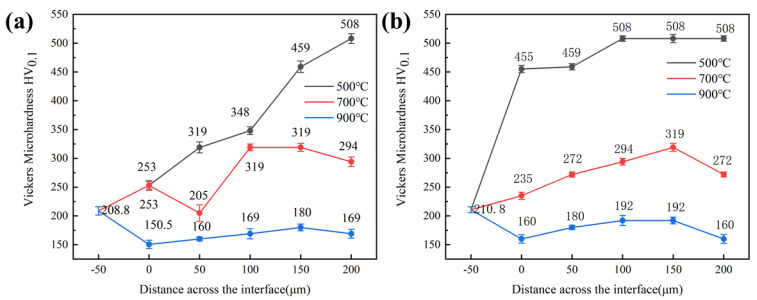
Microhardness–depth curves of CS and HVOF high-entropy alloy coatings heat-treated at different temperatures. (**a**) CS coating; (**b**) HVOF coating.

**Figure 8 materials-16-00055-f008:**
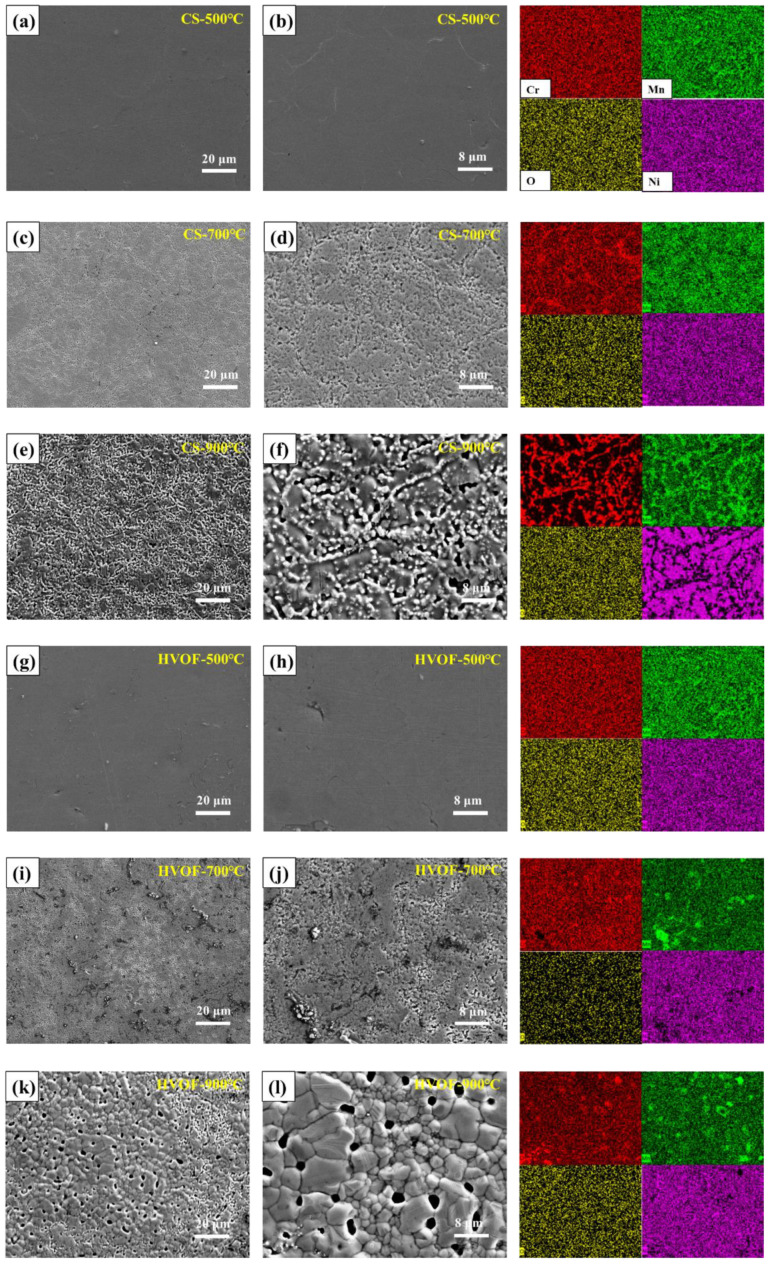
Surface morphology and energy spectrum of CS and HVOF coatings heat-treated at different temperatures. (**a**,**b**) CS-500 °C; (**c**,**d**) CS-700 °C; (**e**,**f**) CS-900 °C; (**g**,**h**) HVOF-500 °C; (**i**,**j**) HVOF-700 °C; (**k**,**l**) HVOF-900 °C.

**Figure 9 materials-16-00055-f009:**
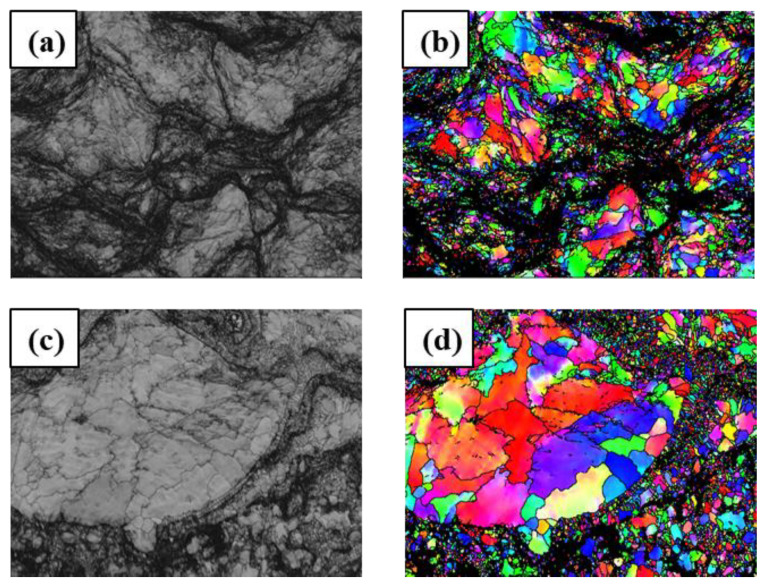
Image quality, phase structure and particle size of electron backscatter diffraction patterns of high-entropy alloy coatings. (**a**,**b**) CS coatings; (**c**,**d**) HVOF coatings.

**Figure 10 materials-16-00055-f010:**
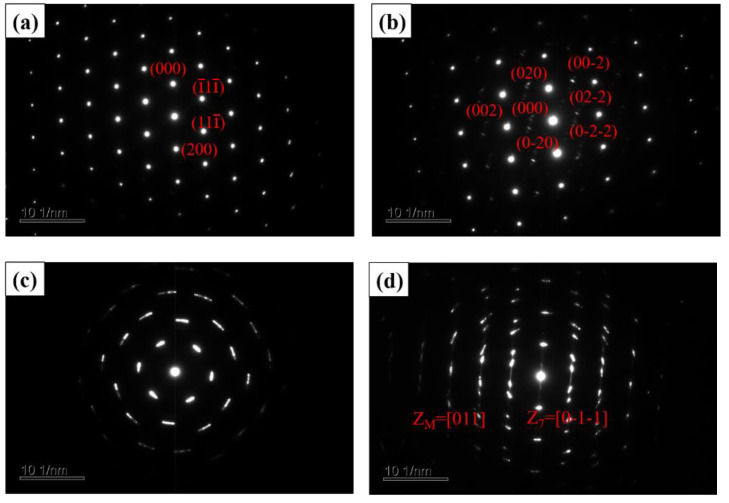
TEM selected area diffraction patterns of high-entropy alloy coatings. (**a**,**b**) CS coatings; (**c**,**d**) HVOF coatings.

**Figure 11 materials-16-00055-f011:**
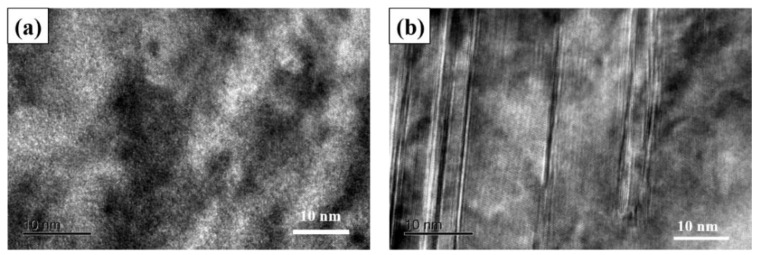
TEM micromorphology of high-entropy alloy coatings. (**a**) CS coating; (**b**) HVOF coating.

**Figure 12 materials-16-00055-f012:**
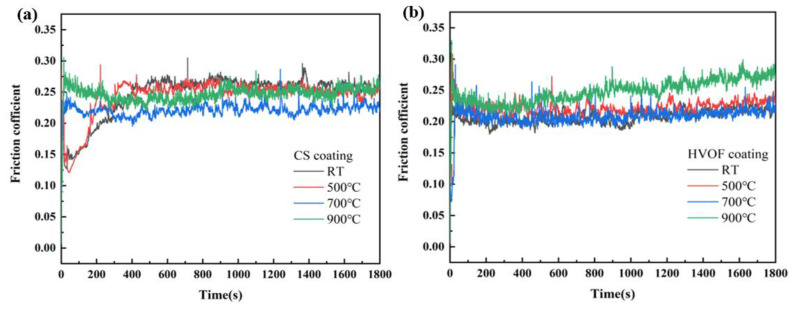
Friction coefficient curves of CoCrFeNiMn high-entropy alloy coatings. (**a**) CS coatings; (**b**) HVOF coatings.

**Figure 13 materials-16-00055-f013:**
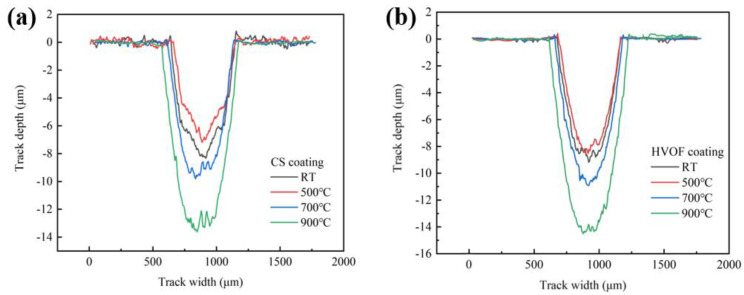
Wear cross-sectional volumes of CoCrFeNiMn high-entropy alloy coatings. (**a**) CS coatings; (**b**) HVOF coatings.

**Figure 14 materials-16-00055-f014:**
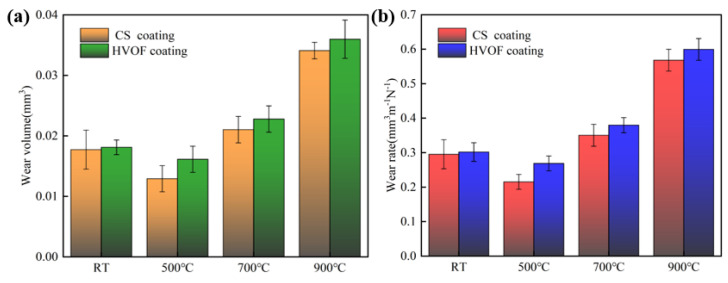
Wear amount and wear rate of CS and HVOF coatings heat-treated at different temperatures. (**a**) Wear amount; (**b**) wear rate.

**Figure 15 materials-16-00055-f015:**
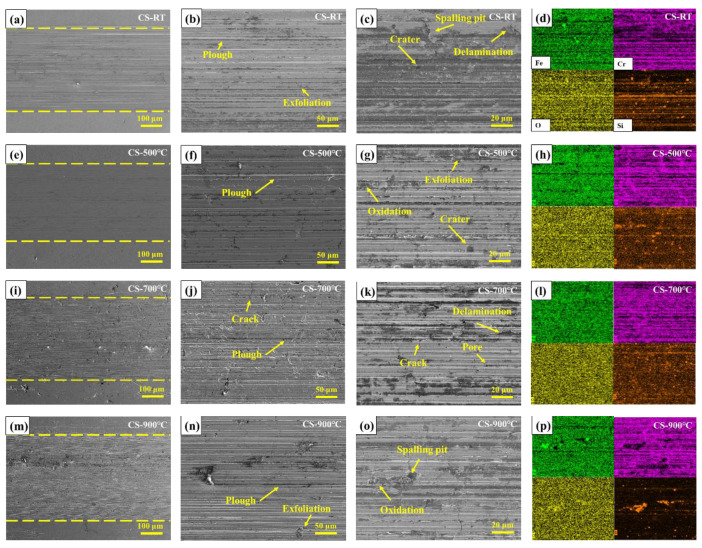
SEM wear morphology and EDS spectrum analysis of CS coatings under different heat-treatment conditions. (**a**–**d**) CS-RT coating; (**e**–**h**) CS-500 °C coating; (**i**–**l**) CS-700 °C coating; (**m**–**p**) CS-900 °C coating.

**Figure 16 materials-16-00055-f016:**
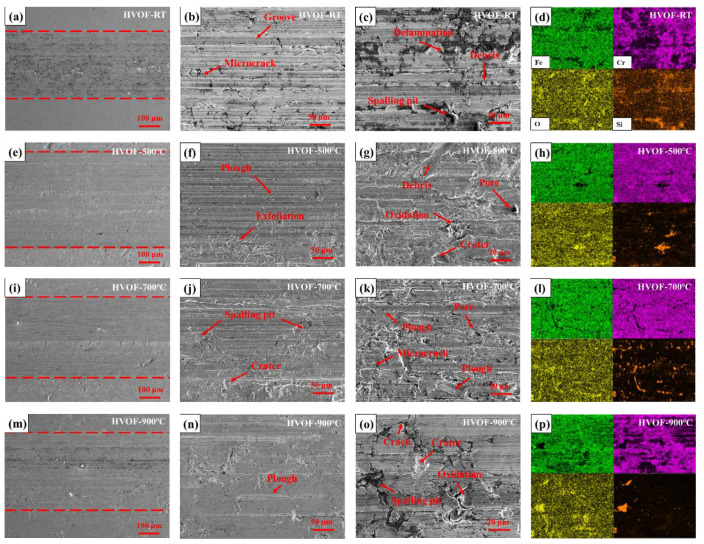
SEM wear morphology and EDS spectrum analysis of HVOF coatings under different heat treatment conditions. (**a**–**d**) HVOF-RT coating; (**e**–**h**) HVOF-500 °C coating; (**i**–**l**) HVOF-700 °C coating; (**m**–**p**) HVOF-900 °C coating.

**Table 1 materials-16-00055-t001:** Specific nominal composition of CoCrFeNiMn high-entropy alloy powders (wt. %).

Fe	Co	Ni	Cr	Mn
20.09	20.96	21.01	18.86	Bal.

**Table 2 materials-16-00055-t002:** Process parameters of CS and HVOF.

CS Process		HVOF Process	
Gas	N_2_	H_2_ Flow rate	10.6 L/s
Pressure	7 MPa	O_2_ Flow rate	3.6 L/s
Temperature	1100 °C	Air Flow rate	5.7 L/s
Standoff distance	15 mm	Standoff distance	250 mm
Robot speed	500 mm/s	Robot speed	500 mm/s
Powder feed	2 rpm	Powder feed	0.5 g/s
Step	1 mm	Step	5 mm
Layers	2	layers	10

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
