# Peer review of "Microstructure and Corrosive Wear Properties of CoCrFeNiMn High-Entropy Alloy Coatings"

_materials, 2022, doi:10.3390/ma16010055_

Round 1

Reviewer 1 Report

The paper titled “Microstructure and corrosive wear properties of CoCrFeNiMn high-entropy alloy coatings” reports the effects of heat treatment temperature (500 °C, 700 °C, 900 °C) on the phase composition of CoCrFeNiMn high-entropy alloy coatings prepared by cold spraying and high velocity oxy-fuel spraying and on their corrosion wear properties. It can be considered for publication after appropriate revision. The comments are listed below.

1.      Line 35-37: “increasing recrystallization temperature, slowing down grain growth, reducing recrystallization temperature,” -> “increasing or reducing recrystallization temperature, slowing down grain growth,”

2.      Captions of all figures should be revised in terms of punctuation.

3.      It is not clear why the XRD patterns for HOVF coating are so different in Fig. 3b and Fig. 4b (500 °C).

4.      Figure 8: Names of the samples and the temperature labels should be written over the images.

5.      Please explain why the wear volume and wear rate after treatment at 500 °C are lower than at RT.

6.      In conclusion, what is “(1)”?

Author Response

We appreciate reviewers for the careful read and thoughtful comments. Additionally, thanks for your evaluation and recommendation for our work. We tried our best to revise and improve the quality of the manuscript according to your meaningful comments. Our point-by-point responses to the reviewer’s comments are as follows.

Reviewer 1:

Comments and Suggestions for Authors

The paper titled “Microstructure and corrosive wear properties of CoCrFeNiMn high-entropy alloy coatings” reports the effects of heat treatment temperature (500 °C, 700 °C, 900 °C) on the phase composition of CoCrFeNiMn high-entropy alloy coatings prepared by cold spraying and high velocity oxy-fuel spraying and on their corrosion wear properties. It can be considered for publication after appropriate revision. The comments are listed below.

  1. Line 35-37: “increasing recrystallization temperature, slowing down grain growth, reducing recrystallization temperature”.

Response: We are sorry for the mistake. The “reducing recrystallization temperature” has been deleted.

  1. Captions of all figures should be revised in terms of punctuation.

Response: We are very sorry for the inconsistent figure captions. The figure captions have been modified to meet the requirements of the journal.

  1. It is not clear why the XRD patterns for HOVF coating are so different in Fig. 3b and Fig. 4b (500 °C).

Response: We really appreciate reviewer for the good question. Materials generally nucleate at low temperature and grow up at high temperature. The 500 °C heat treatment of HVOF alloy coating may be the temperature point at which recrystallization nucleation begins, so some nanocrystals are produced, but the grains do not grow at this temperature, resulting in FCC peak broadening. We have explained in the text.

  1. Figure 8: Names of the samples and the temperature labels should be written over the images.

Response: We really appreciate reviewer for the good advice. The samples and the temperature labels have been written over the figures.

  1. Please explain why the wear volume and wear rate after treatment at 500 °C are lower than at RT.

Response: We really appreciate reviewer for the good question. Comprehensive analysis shows that after heat treatment at 500 °C, the hardness of the coating (HVOF and CS) increases, the internal stress decreases, and the mechanical properties are better, so the wear resistance is the best.

  1. In conclusion, what is “(1)”?

Response: We really sorry for our mistake. The conclusions have been listed in order.

Reviewer 2 Report

The present work reviews the Microstructure and corrosive wear properties of CoCrFeNiMn high-entropy alloy coatings. Overall the work is well-written and outlined. The topic chosen for review is exciting and relevant. This theme will add to the thematic area compared to other publications. The text is clear and easy to read. However, some improvements are needed:

  1. Please ensures that your manuscript meets Materials style requirements.
  2. The abstract part needs to be improved. And the authors need to describe the main results in the abstract. 
  3. The induction part needs to be improved, and the author should describe the result /main finding
  4. the author should explain what the main finding of this paper is 
  5. The author should calculate the lattice parameter for all XRD data to support the author's claim(made in the table) 
  6. The Outlook section should be better structured, and more guidance should be highlighted in the section. 
  7. The conclusion part needs to be improved
  8. Please confirm ensures that your reference format meets Materials style requirements

Author Response

Comments and Suggestions for Authors:

The present work reviews the Microstructure and corrosive wear properties of CoCrFeNiMn high-entropy alloy coatings. Overall the work is well-written and outlined. The topic chosen for review is exciting and relevant. This theme will add to the thematic area compared to other publications. The text is clear and easy to read. However, some improvements are needed:

1.Please ensures that your manuscript meets Materials style requirements.

Response: We thank the reviewer for the good advice. The paper meets Materials style requirements after format modification.

  1. The abstract part needs to be improved. And the authors need to describe the main results in the abstract.

Response: We thank the reviewer for the good suggestion. The abstract part has been improved according to the advice.

  1. The induction part needs to be improved, and the author should describe the result /main finding

Response: We thank the reviewer for the good suggestion. The introduction part has been improved, including adding the number of references, the result /main finding in the literature and adjusting the structure of the introduction part.

  1. the author should explain what the main finding of this paper is

Response: We really appreciate reviewer for the nice advice. The main finding of the paper has been added as follows: “Although there have been many reports on the wear and mechanical properties of high-entropy alloys after heat treatment, there are few studies on the wear behavior of CoCrFeNiMn high-entropy alloy coating after heat treatment, especially under simulated seawater drilling fluid.”

  1. The author should calculate the lattice parameter for all XRD data to support the author's claim (made in the table).

We apologize for the loose interpretation. Due to the different degree of lattice distortion of each coating (different spraying process and different heat treatment temperature), the lattice parameters are difficult to be specifically determined, but can be reflected in the XRD pattern.

  1. The Outlook section should be better structured, and more guidance should be highlighted in the section.

Response: We thank the reviewer for the good suggestion. We have arranged the structure of the outlook section and added more references as follows:

[1] Hou, J.L.; Li, Q.; Wu, C.B.; Zheng, L.M. Atomic simulations of grain structures and deformation behaviors in nanocrystalline cocrfenimn high-entropy alloy. Materials. 2019, 12(7), 1010. https://doi.org/10.3390/ma12071010.

[2] Stepanov, N.D.; Shaysultanov, D.G.; Yurchenko, N.Yu.; Zherebtsov, S.V.; Ladygin, A.N.; Salishchev, G.A.; Tikhonovsky, M.A. High temperature deformation behavior and dynamic recrystallization in CoCrFeNiMn high entropy alloy. Mater. Sci. Eng. A-Struct. Mater. Prop. Microstruct. Process. 2015, 636, 188-195. https://doi.org/10.1016/j.msea.2015.03.097.

[3] Akisin, C.J.; Bennett, C.J.); Venturi, F.; Assadi, H.; Hussain, T. Numerical and experimental analysis of the deformation behavior of CoCrFeNiMn high entropy alloy particles onto various substrates during cold spraying. J. Therm. Spray Technol. 2022, 31(4), 1085-1111. https://doi.org/10.1007/s11666-022-01377-1.

[4] Feng, S.; Guan, S.; Story, W.A.; Ren, J.; Zhang, S.B. Cold spray additive manufacturing of CoCrFeNiMn high-entropy alloy: Process development, microstructure, and mechanical properties. J. Therm. Spray Technol. 2022, 31(4), 1222-1231. https://doi.org/ 10.1007/s11666-022-01374-4.

  1. The conclusion part needs to be improved.

Response: We thank the reviewer for the good suggestion. The conclusion part has been improved as follows:

The CoCrFeNiMn high-entropy alloy coatings were prepared by cold spraying and high velocity oxy-fuel spraying. The effects of heat treatment on the phase composition, microhardness and wear behavior of high-entropy alloy coatings were studied. The main results are as follows:

(1) CoCrFeNiMn high-entropy alloy powders prepared by gas atomization for CS and HVOF have good spherical shape, smooth surface and dendritic structure. The CoCrFeNiMn high-entropy alloy coatings prepared by CS and HVOF have dense structure and bond well with the substrate. They are all single FCC solid solution structure, and the porosity is less than 1.5 %. Compared with CS coating, HVOF coating has higher porosity.

(2) After heat treatment, the main peaks of all oriented FCC phase are broadened and the strength is obviously enhanced. Both coatings reach the maximum hardness after vacuum heat treatment at 500 °C, and the Vickers microhardness of CS-500 °C and HVOF-500 °C are 487.6 and 352.4 HV0.1, respectively.

(3) The wear rates of the two coatings at room temperature are very close. The wear rate for the CS and HVOF coatings achieves the lowest after vacuum heat treatment at 500 ° C. The wear rate of CS-500 °C coating is 0.2152 mm3m-1N-1, which is about 4/5 (0.2651 mm3m-1N-1) of that of HVOF-500 °C coating. The CS coating with 500 ℃ vacuum heat treatment has the best wear resistance due to the highest microhardness. The wear rates and wear amounts of the two coatings heat treated at 700 °C and 900 °C decreases due to the decrease of microhardness. The wear mechanisms of CS and HVOF coatings before and after vacuum heat treatment are adhesive wear, abrasive wear, fatigue wear and oxidation wear.

  1. Please confirm ensures that your reference format meets Materials style requirements.

Response: We thank the reviewer for the good advice. The reference format meets Materials style requirements.

Reviewer 3 Report

The work is devoted to the study of corrosion resistance and wear resistance of a high-entropy alloy for use in aggressive environments. Although the topic is quite interesting, the authors did not show the significance of their work. The introduction is inconsistent and does not provide values ​​for the properties of materials already studied. At the end of the manuscript, the authors do not compare their results with the results of other works. Authors should also read the stylistic requirements for publication in MDPI journals very carefully.

The manuscript requires serious improvements, and in its current form cannot be accepted. After correcting all the comments, it is necessary to re-review the work.

1. Introduction

            Lines 39-40. “CoCrFeNi alloys with face-centered cubic structure have different microstructures and mechanical properties.– Authors should expand this paragraph significantly and cite specific publications to show the difference in microstructures they are talking about, as well as provide numerical values for the properties of these materials.

            Lines 40-41 “As one of the simplest structures, CoCrFeNiMn alloys with face-centered cubic structure have been widely studied [6,7]– Authors should provide more details. The use of the word "widely" in this case seems somewhat strange, since the authors provided only 2 references.

            Lines 41-42 “Heat treatment promotes the uniformity of phase and grain size in the alloy without changing the proportion of the original elements, thereby improving mechanical properties [8-10]– The authors should again provide more details. Heat treatment under what conditions? Give comparisons with works where there was no heat treatment. Give numerical values of mechanical properties in the format "[PROPERTY] increased from [VALUE] to [VALUE]"

            Lines 43-44 “High velocity oxygen fuel spraying technology has higher particle velocity, lower flame temperature and 43 less unmelted particles, which can prepare anticorrosive and wear-resistant coatings with low oxidation degree, 44 good bonding rate and low porosity [11,12]– The previous paragraph and the current one have nothing to do with each other. “spraying technology has higher particle velocity” – Has higher features in comparison with what?

Lines 54-56. Why do twins improve the properties of materials?

Lines 56-61 “Both of them show good wear resistance and corrosion resistance.” – Authors should again give the numerical values of the properties, in this case, corrosion resistance and wear resistance.

Lines 62-64 “Although there have been many reports on the room temperature wear and mechanical properties of 62 CoCrFeNiMn high-entropy alloy coatings after heat treatment, there are few studies on the corrosion wear 63 behavior after heat treatment – Authors should give the references.

Summary: The introduction contains insufficient information to judge the relevance of the work. The introduction should be seriously supplemented. In particular, the authors state that there are few studies on corrosion wear of coatings after heat treatment. But do they exist? Accordingly, authors should present the already reported results and be more specific about the problem. Authors should give articles on the study of corrosion resistance of coatings without heat treatment and provide numerical values.

2. Experimental       

            Lines 70-71 “In this study, high purity metal materials, high purity iron particles, cobalt particles, chromium particles, 70 nickel particles, and manganese particles (purity > 99.99 %) were selected as raw material powders.– Why do the authors write "high-purity metal materials" if they list them further? Delete the phrase "high-purity metal materials".

            Authors should indicate the manufacturers of the powders and their particle size before the atomization process.

            The 71 equiatomic CoCrFeNiMn high-entropy alloy powders were prepared by vacuum atomization with high 72 pressure nitrogen as the atmosphere medium.– Authors should give more details on the atomization process.

            Table 1 – What does “Mn - Bal” mean?

            Lines 77-78 “Spray drying was performed for the high-entropy alloy powders to allow powder particles to 77 flow freely in the spray gun.– Authors should rewrite this sentence.

            Line 78 “The substate material is 35CrMo alloy structural steel– Tenses are inconsistent in the paragraph. Everywhere the authors used the past tense, and in this phrase the present.

            Lines 80-82 “In order to obtain 80 better mechanical bonding strength between the coating and the substrate, and appropriate surface roughness, 81 the pretreatment before spraying is to degrease and sandblast the 35CrMo substrate.Authors should rewrite this sentence.

            Lines 86-87 „were used to spraying coatings.“ – after „to“ should be infinitive

            Table 2 the word "layers" should be capitalized like the rest of the words in the table.

            Lines 125-126 “The load was 125 100 gf and the duration was 10 s.” – Authors should give the load in kg or N.

3. Results and discussion

            Line 136-137 “The enlarged SEM secondary electron images show that there are dendrites on the surface of the high-entropy alloy powders...– Where can the readers see this enlarged Figure and the dendritic structure?

            Figure 2. Do these peaks belong to the high-entropy phase? Can the authors give structural parameters of the obtained phase in the reference and refined state using the Rietveld method?

             Figure 3. Why are there no substrate peaks in the diffraction pattern? What phase does the peak in the range from 93 to 100 degrees belong to?

            Figure 4. and Line 166 “This is mainly due to the grain refinement, lattice constant increase, unfavorable crystallinity and 166 lattice distortion caused by heat treatment– Authors should provide crystal lattice parameters to discuss about lattice constants.

            Figure 4b. Why, before annealing, the coating has a crystalline structure (Figure 3b), and after annealing at a temperature of 500 °Ð¡, the authors obtained an X-ray amorphous structure. At the same time, at 700 °Ð¡, the authors again obtained a crystalline structure.

            Line 177 “and the FCC phase of the CoCrFeNiMn high-entropy alloy is easily deformed– How did the authors come to this conclusion?

            Figure 5a and c. The substrate is marked at the bottom of the images, the coating is in the middle, and what kind of microstructure is shown at the top of the image (having black color)?

            Figure 7. The authors should once again check the HOVF XRD pattern (Figure 4b) of the coating after annealing at 500 °Ð¡. It is strange that with an X-ray amorphous structure, the coating also has the maximum hardness.

            Figure 8. Authors should provide mapping (colored right images) with a higher quality. There is absolutely nothing to see in the pictures.

            Figure 8 l. Are the black spots on the SEM image dendritic cores?

            Lines 235-242 “The relatively small grain size of CS coating may be due to the excessive movement and interaction 235 of dislocations during dynamic loading. CS treatment has obvious refining effect on the grains of 236 CoCrFeNiMn high-entropy alloy, and the particles seriously deformed due to dynamic inertia effect and 237 adiabatic deformation during impact. At the same time of high strain plastic deformation, the dislocations at 238 the crystal interface increase. The dislocation density also increases to form dislocation cells, and the coarse 239 grains are refined into subgrains. In the local interface region where the maximum plastic deformation occurs 240 in the granular material, dynamic recrystallization occurs under the combined action of adiabatic heating and 241 plastic deformation, which further refines the subgrains into ultrafine grains – The authors should give references to prove it.

            Figure 9 c and f. Authors should provide the images with a higher quality. There is absolutely nothing to see in the pictures. Authors should also provide numerical values of grain sizes in the text.

            Lines 273-274 “The friction coefficients of RT and 500 °C heat treated CS coatings 273 are relatively low in the first 200 s, which is due to the fact that the oxide film on the CS coating is relatively 274 dense” – What kind of oxide film?

            Figure 12. What is the friction coefficient of the substrate without coatings?

            Figure 15 d, h, l, p – Authors should provide the images with a higher quality. There is absolutely nothing to see in the pictures.

            Figure 16. Authors should provide the images with a higher quality. There is absolutely nothing to see in the pictures.

            Summarize the results why the coatings obtained at 500 °Ð¡ annealing have the highest mechanical properties.

            Since your research is about solving the problem of drill pipes operating in corrosive environments, show the advantages and disadvantages of your coatings compared to other coatings. Make a table with the hardness, wear resistance and corrosion resistance of your coating, other coatings and original substrates.

4. Conclusions

            Lines 371-375 “Both coatings reach the maximum hardness after vacuum heat treatment at 500 °C, and the Vickers microhardness of CS-500 °C is 487.6 HV. The wear rates of the two coatings at room temperature are very close. The wear rate for the CS and HVOF coatings achieves the lowest after vacuum heat treatment at 500 ° C. The wear rate of CS-500 °C coating is 0.2152 mm3m-1N -1 , which is about 4/5 of that of HVOF-500 °C coating”. – Briefly describe why at 500 °C you achieved the highest properties and give their values in one sentence.

            Lines 375-377 “The wear rates and wear amounts of the two coatings heat treated at 700 °C and 900 °C decrease with the increase of heat treatment temperature.” – Briefly describe why?

Abstract

            The abstract should grab the reader's attention, so it should be completely rewritten. Indicate in the annotation the problem of corrosion of drill pipes. Write about how your research suggests one of the solutions to this problem by obtaining high-entropy coatings. Very briefly describe the methods by which you obtained the coatings. Indicate that as a result you got dense single-phase coatings, while annealing at a temperature of 500 °C allows you to achieve maximum properties (give the values). Describe the wear mechanism. Ideally, give a brief comparison of your coatings with analogues and how they turned out to be better or worse.

Author Response

Comments and Suggestions for Authors

The work is devoted to the study of corrosion resistance and wear resistance of a high-entropy alloy for use in aggressive environments. Although the topic is quite interesting, the authors did not show the significance of their work. The introduction is inconsistent and does not provide values for the properties of materials already studied. At the end of the manuscript, the authors do not compare their results with the results of other works. Authors should also read the stylistic requirements for publication in MDPI journals very carefully. The manuscript requires serious improvements, and in its current form cannot be accepted. After correcting all the comments, it is necessary to re-review the work.

  1. Introduction

Lines 39-40. “CoCrFeNi alloys with face-centered cubic structure have different microstructures and mechanical properties.” – Authors should expand this paragraph significantly and cite specific publications to show the difference in microstructures they are talking about, as well as provide numerical values for the properties of these materials.

Response: We thank the reviewer for the good suggestion. The contents of this section have been redescribed as follows: “The atomic radii of Co, Cr, Fe and Ni are 0.126, 0.127, 0.127, 0.124 nm respectively. They are easy to form FCC structure, and CoCrFeNi-based alloys are common combinations in high entropy alloys.”

Lines 40-41 “As one of the simplest structures, CoCrFeNiMn alloys with face-centered cubic structure have been widely studied [6,7]” – Authors should provide more details. The use of the word "widely" in this case seems somewhat strange, since the authors provided only 2 references.

Response: We thank the reviewer for the good advice. The examples of CoCrFeNiMn high entropy alloys were added later.

Lines 41-42 “Heat treatment promotes the uniformity of phase and grain size in the alloy without changing the proportion of the original elements, thereby improving mechanical properties [8-10]” – The authors should again provide more details. Heat treatment under what conditions? Give comparisons with works where there was no heat treatment. Give numerical values of mechanical properties.

Response: We thank the reviewer for the good advice. The details of heat treatment on high entropy alloy were added later.

Lines 43-44 “High velocity oxygen fuel spraying technology has higher particle velocity, lower flame temperature and less unmelted particles, which can prepare anticorrosive and wear-resistant coatings with low oxidation degree, good bonding rate and low porosity [11,12]” – The previous paragraph and the current one have nothing to do with each other. “spraying technology has higher particle velocity” – Has higher features in comparison with what?

Response: We thank the reviewer for the good question. The sentence has been revised as “High velocity oxygen fuel spraying technology has higher particle velocity, lower flame temperature and less unmelted particles than atmospheric plasma spraying”.

Lines 54-56. Why do twins improve the properties of materials?

Response: We really appreciate reviewer for the good question. Generally, crystals with low stacking fault energy are prone to produce twins, which can reduce plasticity and play a certain role in hardening, thus improving their comprehensive mechanical properties.

Lines 56-61 “Both of them show good wear resistance and corrosion resistance.”-Authors should again give the numerical values of the properties, in this case, corrosion resistance and wear resistance.

Response: We thank the reviewer for the good advice. The the numerical values has been added as follows: “The corrosion current and wear rate of the CS coating were 1.2 μA and 2.80Í10-4 mm3N-1m-1. Besides, the corrosion current and wear rate of the HVOF coating were 0.29 μA and 6.70Í10-5 mm3N-1m-1. Both of them showed good wear resistance and corrosion resistance.”

Lines 62-64 “Although there have been many reports on the room temperature wear and mechanical properties of CoCrFeNiMn high-entropy alloy coatings after heat treatment, there are few studies on the corrosion wear behavior after heat treatment” – Authors should give the references.

Response: We thank the reviewer for the good advice. The relevant literature has been mentioned above.

Summary: The introduction contains insufficient information to judge the relevance of the work. The introduction should be seriously supplemented. In particular, the authors state that there are few studies on corrosion wear of coatings after heat treatment. But do they exist? Accordingly, authors should present the already reported results and be more specific about the problem. Authors should give articles on the study of corrosion resistance of coatings without heat treatment and provide numerical values.

Response: We thank the reviewer for the good advice. We have made additions to the introduction, such as the influence of heat treatment on the mechanical properties and wear properties of high entropy alloys, and redesigned the structure of the introduction.

  1. Experimental

Lines 70-71 “In this study, high purity metal materials, high purity iron particles, cobalt particles, chromium particles, nickel particles, and manganese particles (purity > 99.99 %) were selected as raw material powders.” – Why do the authors write "high-purity metal materials" if they list them further? Delete the phrase "high-purity metal materials". Authors should indicate the manufacturers of the powders and their particle size before the atomization process.

Response: We really sorry for our mistake. The raw materials are not powders, but block metals. The relevant description has been revised as “high purity bulk iron, cobalt, chromium, nickel, and manganese (purity > 99.99 %, produced by BGRIMM Technology group) were selected as raw material”.

“The equiatomic CoCrFeNiMn high-entropy alloy powders were prepared by vacuum atomization with high pressure nitrogen as the atmosphere medium.” Authors should give more details on the atomization process.

Response: We thank the reviewer for the good advice. The corresponding description has been modified as “The bulk metal was heated in a crucible according to a certain atomic ratio, and N2 was used to protect the melting process. The molten metal liquid was formed into fine liquid particles by atomizing nozzle under vacuum environment and cooled to obtain high entropy alloy powders.”

Table 1 – What does “Mn - Bal” mean?

Response: We are glad to explain to you. The “Bal.” means “Balanced”, which refers to the remaining proportion.

Lines 77-78 “Spray drying was performed for the high-entropy alloy powders to allow powder particles to flow freely in the spray gun.” – Authors should rewrite this sentence.

Response: We thank the reviewer for the good advice. The sentence has been deleted.

Line 78 “The substate material is 35CrMo alloy structural steel” – Tenses are inconsistent in the paragraph. Everywhere the authors used the past tense, and in this phrase the present.

Response: We thank the reviewer for the good advice. The “is” has been revised as “was”.

Lines 80-82 “In order to obtain better mechanical bonding strength between the coating and the substrate, and appropriate surface roughness, the pretreatment before spraying is to degrease and sandblast the 35CrMo substrate.” – Authors should rewrite this sentence.

Response: We thank the reviewer for the good advice. The sentence has been revised as “The 35CrMo substrate was degreased and sandblasted in order to obtain better mechanical bonding strength and appropriate surface roughness”.

Lines 86-87 were used to spraying coatings.

Response: We thank the reviewer for the good advice. The sentence has been revised as “were used to prepare coatings”.

Table 2 the word "layers" should be capitalized like the rest of the words in the table.

Response: We really sorry for our mistake. The "layers" has been revised as “Layers”.

Lines 125-126 “The load was 100 gf and the duration was 10 s.” – Authors should give the load in kg or N.

Response: We thank the reviewer for the good suggestion. The sentence has been revised as “The load was 1 N and the duration was 10 s.”

  1. Results and discussion

Line 136-137 “The enlarged SEM secondary electron images show that there are dendrites on the surface of the high-entropy alloy powders...” – Where can the readers see this enlarged Figure and the dendritic structure?

Response: We are sorry for the unclear description. The previous enlarged SEM secondary electron images were too small to see the microscopic structure of the powder clearly. This phenomenon can be found by using the enlarged SEM secondary electron images as Fig.1(c) and (d).

Figure 2. Do these peaks belong to the high-entropy phase? Can the authors give structural parameters of the obtained phase in the reference and refined state using the Rietveld method?

Response: We thank the reviewer for the good question. The high entropy alloy phase refers to the solid solution phase of a variety of elements, with no or little precipitated phase compared with conventional alloys. The peaks in Figure 2 is the FCC phase, which is belong to the high-entropy phase.

Figure 3. Why are there no substrate peaks in the diffraction pattern? What phase does the peak in the range from 93 to 100 degrees belong to?

Response: We thank the reviewer for the good question. There are no substrate peaks in the diffraction pattern because the coating is too thick (250-300 μm). Generally, XRD does not need to show the phases after 90 degrees, so it is not marked in the Fig.3.

Figure 4. and Line 166 “This is mainly due to the grain refinement, lattice constant increase, unfavorable crystallinity and lattice distortion caused by heat treatment” – Authors should provide crystal lattice parameters to discuss about lattice constants.

Response: We thank the reviewer for the good suggestion. Lattice constant additions are not appropriate here, so the “lattice constant increase” have been deleted from the paper.

Figure 4b. Why, before annealing, the coating has a crystalline structure (Figure 3b), and after annealing at a temperature of 500 °Ð¡, the authors obtained an X-ray amorphous structure. At the same time, at 700 °Ð¡, the authors again obtained a crystalline structure.

Response: We thank the reviewer for the good question. Materials generally nucleate at low temperature and grow up at high temperature. The 500 °C heat treatment of HVOF alloy coating may be the temperature point at which recrystallization nucleation begins, so some nanocrystals are produced, but the grains do not grow at this temperature, resulting in FCC peak broadening. When the heat treatment temperature exceeds 500℃, the recrystallization behavior of the high entropy alloy results in the increase of grain size and the reappearance of crystalline structure.

Line 177 “and the FCC phase of the CoCrFeNiMn high-entropy alloy is easily deformed” – How did the authors come to this conclusion?

Response: We thank the reviewer for the good question. There are studies to prove this, such as the following references:

[1] Hou, J.L.; Li, Q.; Wu, C.B.; Zheng, L.M. Atomic simulations of grain structures and deformation behaviors in nanocrystalline cocrfenimn high-entropy alloy. Materials. 2019, 12(7), 1010. https://doi.org/10.3390/ma12071010.

[2] Stepanov, N.D.; Shaysultanov, D.G.; Yurchenko, N.Yu.; Zherebtsov, S.V.; Ladygin, A.N.; Salishchev, G.A.; Tikhonovsky, M.A. High temperature deformation behavior and dynamic recrystallization in CoCrFeNiMn high entropy alloy. Mater. Sci. Eng. A-Struct. Mater. Prop. Microstruct. Process. 2015, 636, 188-195. https://doi.org/10.1016/j.msea.2015.03.097.

Figure 5a and c. The substrate is marked at the bottom of the images, the coating is in the middle, and what kind of microstructure is shown at the top of the image (having black color)?

Response: We thank the reviewer for the good question. The image (having black color) is the filler, and the coating section is in the middle of the filler.

Figure 7. The authors should once again check the HOVF XRD pattern (Figure 4b) of the coating after annealing at 500 °Ð¡. It is strange that with an X-ray amorphous structure, the coating also has the maximum hardness.

Response: We thank the reviewer for the good question. It is also proved that amorphous alloys have high hardness: [3] Shu, F.Y.; Wang, B.; Zhao, H.Y.; Tan, C.W.; Zhou, J.L.; Zhang, J. Effects of line energy on microstructure and mechanical properties of CoCrFeNiBSi high-entropy alloy laser cladding coatings. J. Therm. Spray Technol. 2020, 29(4), 789-797. https://doi.org/10.1007/s11666-020-01004-x.

Figure 8. Authors should provide mapping (colored right images) with a higher quality. There is absolutely nothing to see in the pictures.

Response: We thank the reviewer for the good advice. The EDS mapping of the coatings (colored right images) is obtained based on the middle images. It is true that the pixel of the image is low, but we will label each element of the EDS mapping image, which we think can reveal some phenomena.

Figure 8 l. Are the black spots on the SEM image dendritic cores?

Response: We thank the reviewer for the good question. Yes, the black spots on the SEM image are dendritic cores.

Lines 235-242 “The relatively small grain size of CS coating may be due to the excessive movement and interaction of dislocations during dynamic loading. CS treatment has obvious refining effect on the grains of 236 CoCrFeNiMn high-entropy alloy, and the particles seriously deformed due to dynamic inertia effect and 237 adiabatic deformation during impact. At the same time of high strain plastic deformation, the dislocations at 238 the crystal interface increase. The dislocation density also increases to form dislocation cells, and the coarse 239 grains are refined into subgrains. In the local interface region where the maximum plastic deformation occurs 240 in the granular material, dynamic recrystallization occurs under the combined action of adiabatic heating and 241 plastic deformation, which further refines the subgrains into ultrafine grains” – The authors should give references to prove it.

Response: We thank the reviewer for the good advice. The corresponding literature has been cited as follows:

[4] Akisin, C.J.; Bennett, C.J.); Venturi, F.; Assadi, H.; Hussain, T. Numerical and experimental analysis of the deformation behavior of CoCrFeNiMn high entropy alloy particles onto various substrates during cold spraying. J. Therm. Spray Technol. 2022, 31(4), 1085-1111. https://doi.org/10.1007/s11666-022-01377-1.

[5] Feng, S.; Guan, S.; Story, W.A.; Ren, J.; Zhang, S.B. Cold spray additive manufacturing of CoCrFeNiMn high-entropy alloy: Process development, microstructure, and mechanical properties. J. Therm. Spray Technol. 2022, 31(4), 1222-1231. https://doi.org/ 10.1007/s11666-022-01374-4.

Figure 9 c and f. Authors should provide the images with a higher quality. There is absolutely nothing to see in the pictures. Authors should also provide numerical values of grain sizes in the text.

Response: We are sorry for the unclear picture. Given that other figures can also reflect this phenomenon, the Figure 9 c and f has been deleted.

Lines 273-274 “The friction coefficients of RT and 500 °C heat treated CS coatings are relatively low in the first 200 s, which is due to the fact that the oxide film on the CS coating is relatively dense” – What kind of oxide film?

Response: We thank the reviewer for the good advice. The oxide films should be Cr2O3 and CoO by consulting references:

[6] Cui, Y.; Shen, J.Q.; Manladan, S.M.; Geng, K.P.; Hu, S.S. Effect of heat treatment on the FeCoCrNiMnAl high-entropy alloy cladding layer. Surf. Eng. 2021, 37(12), 1532-1540. https://doi.org/10.1080/02670844.2021.1952048.

[7] Silvello, A.; Cavaliere, P.; Yin, S.; Lupoi, R.; Cano, I.G.; Dosta, S. Microstructural, mechanical and wear behavior of HVOF and Cold-sprayed high-entropy alloys (HEAs) coatings. J. Therm. Spray Technol. 2022, 31(4) 1184-1206. https://doi.org/10.1007/s11666-021-01293-w.

Figure 12. What is the friction coefficient of the substrate without coatings?

Response: We thank the reviewer for the good advice. The friction coefficient of the substrate is about 0.35, which is higher than that of the coatings.

Figure 15 d, h, l, p – Authors should provide the images with a higher quality. There is absolutely nothing to see in the pictures.

Response: We thank the reviewer for the good advice. The EDS mapping of the coatings (colored right images) is obtained based on the third image per row. It is true that the pixel of the image is low, but we will label each element of the EDS mapping image, which we think can reveal some phenomena.

Figure 16. Authors should provide the images with a higher quality. There is absolutely nothing to see in the pictures.

Response: We thank the reviewer for the good advice. The EDS mapping of the coatings (colored right images) is obtained based on the third image per row. It is true that the pixel of the image is low, but we will label each element of the EDS mapping image, which we think can reveal some phenomena.

Summarize the results why the coatings obtained at 500 °Ð¡ annealing have the highest mechanical properties.

Response: We thank the reviewer for the good question. The heat treatment at 500 °C makes the coating form a heterogeneous structure composed of nano-twins in the recrystallized grains, and high-density dislocations appear in the deformation region. When the annealing temperature exceeds 500 °C, the coating exhibits a completely recrystallized structure, the yield stress decreases, and the hardness decreases.

Since your research is about solving the problem of drill pipes operating in corrosive environments, show the advantages and disadvantages of your coatings compared to other coatings. Make a table with the hardness, wear resistance and corrosion resistance of your coating, other coatings and original substrates.

Response: We thank the reviewer for the good advice. There are few studies on surface protection of drilling tools with high entropy alloy. Our study explores the application potential of high entropy alloy in the field of drilling tool protection. Therefore, it is difficult to provide a comparison of performance with other coatings in the field of drilling tool protection. However, we provide a comparison with the original substrate, as described below:

The wear volume and wear rate of the substrate are 0.0275 mm3和0.4534 mm3m-1N-1, respectively. This also shows that the coatings can protect the substrate after reasonable heat treatment.

  1. Conclusions

Lines 371-375 “Both coatings reach the maximum hardness after vacuum heat treatment at 500 °C, and the Vickers microhardness of CS-500 °C is 487.6 HV. The wear rates of the two coatings at room temperature are very close. The wear rate for the CS and HVOF coatings achieves the lowest after vacuum heat treatment at 500 ° C. The wear rate of CS-500 °C coating is 0.2152 mm3m-1N -1 , which is about 4/5 of that of HVOF-500 °C coating”. – Briefly describe why at 500 °C you achieved the highest properties and give their values in one sentence.

Response: We thank the reviewer for the good advice. The sentence has been revised as: The wear rate for the CS and HVOF coatings achieves the lowest after vacuum heat treatment at 500 ° C. The wear rate of CS-500 °C coating is 0.2152 mm3m-1N-1, which is about 4/5 (0.2651 mm3m-1N-1) of that of HVOF-500 °C coating. The CS coating with 500 ℃ vacuum heat treatment has the best wear resistance due to the highest microhardness.

Lines 375-377 “The wear rates and wear amounts of the two coatings heat treated at 700 °C and 900 °C decrease with the increase of heat treatment temperature.” – Briefly describe why?

Response: We thank the reviewer for the good question. High temperature heat treatment (700, 900 °C) leads to grain coarsening, thus reducing hardness and wear resistance. The corresponding description has been modified.

Abstract

The abstract should grab the reader's attention, so it should be completely rewritten. Indicate in the annotation the problem of corrosion of drill pipes. Write about how your research suggests one of the solutions to this problem by obtaining high-entropy coatings. Very briefly describe the methods by which you obtained the coatings. Indicate that as a result you got dense single-phase coatings, while annealing at a temperature of 500 °C allows you to achieve maximum properties (give the values). Describe the wear mechanism. Ideally, give a brief comparison of your coatings with analogues and how they turned out to be better or worse.

Response: We thank the reviewer for the good advice. The abstract part has been rewritten as follows:

In order to improve the wear resistance of offshore drilling equipment, the CoCrFeNiMn high en-tropy alloy coatings were prepared by cold spraying (CS) and high speed oxygen fuel spraying (HVOF), and the coatings were subjected to vacuum heat treatment at different temperatures (500°C, 700°C and 900°C). The friction and wear experiments of the coatings before and after vacuum heat treatment were carried out in simulated seawater drilling fluid. The results show that CoCrFeNiMn high-entropy alloy coatings prepared by CS and HVOF have dense structure and bond well with the substrate. After vacuum heat treatment, the main peaks of all oriented FCC phases are broadened and the peak strength is obviously enhanced. The two types of coat-ings achieve maximum hardness after vacuum heat treatment at 500 ° C, the Vickers microhard-ness of CS-500 °C and HVOF-500 °C are 487.6 and 352.4 HV0.1, respectively. The wear rates of the two coatings at room temperature are very close. CS and HVOF coatings both have the lowest wear rate after vacuum heat treatment at 500 °C. The CS-500 °C coating has the lowest wear rate of 0.2152 mm3m-1N-1, about 4/5(0.2651 mm3m-1N-1) of the HVOF-500 °C coating. The wear rates and wear amounts of the two coatings heat treated at 700 °C and 900 °C decrease due to the de-crease of microhardness. The wear mechanisms of the coatings before and after vacuum heat treatment are adhesive wear, abrasive wear, fatigue wear and oxidation wear.

Round 2

Reviewer 1 Report

The manuscript was revised thoroughly. Most of the comments are addressed. There are two more comments, which can be reflected during the proofreading step. Therefore, I recommend this paper for publication.

1.      Regarding the XRD patterns for HOVF coating, I did not understand the explanation. What is the treatment temperature for the samples, which patterns are presented in Fig. 2 and 3? If this temperature does not exceed 500 °C, then the patterns in Fig. 3b and Fig. 4b (500 °C) should be the same. If not, please, note the temperature in the captions of Fig. 2 and 3.

2.      Some Chinese symbols appeared in line 341.

Author Response

Detailed responds to reviewers’ comments:

We appreciate reviewers for the careful read and thoughtful comments. Additionally, thanks for your evaluation and recommendation for our work. We tried our best to revise and improve the quality of the manuscript according to your meaningful comments. Our point-by-point responses to the reviewer’s comments are as follows.

Reviewer 1:

Comments and Suggestions for Authors

The manuscript was revised thoroughly. Most of the comments are addressed. There are two more comments, which can be reflected during the proofreading step. Therefore, I recommend this paper for publication.

  1. Regarding the XRD patterns for HOVF coating, I did not understand the explanation. What is the treatment temperature for the samples, which patterns are presented in Fig. 2 and 3? If this temperature does not exceed 500 °C, then the patterns in Fig. 3b and Fig. 4b (500 °C) should be the same. If not, please, note the temperature in the captions of Fig. 2 and 3.

Response: We really appreciate reviewer for the good question. Figure 2 shows the XRD patterns of high-entropy alloy powders, and Figure 3 shows the XRD patterns of CoCrFeNiMn high-entropy alloy coatings without heat treatment. Besides, Figure 4 shows the XRD patterns of CoCrFeNiMn high-entropy alloy coatings after heat treatment at different temperatures. In the process of HVOF, the powders are shot to the substrate at a high speed through a high-pressure spray gun, and the temperature in the spray gun is about 2200-2500 ℃; In the process of CS, the powders are sprayed on the substrate by gas at low temperature and high pressure, and the temperature is about 400-500 ℃. For both HVOF and CS, the retention time of the powders in the spray gun is very short, and the phase is basically unchanged, so the phase structures in Figure 3 and Figure 2 are basically the same. The phase structures of the coatings may change after heat treatment at different temperatures. In addition, we have marked them in the figures in order to better distinguish Figure 2-4.

  1. Some Chinese symbols appeared in line 341.

Response: We really sorry for our mistake. The relevant syntax errors have been corrected.

Reviewer 3 Report

The authors did a great job and responded to all comments. I recommend the article for publication.

Author Response

We appreciate reviewers for the careful read and thoughtful comments. Additionally, thank you for your recognition of our work.